# KidneyCare Guided Immuno-Optimization in Renal Allografts: The KIRA Protocol

**DOI:** 10.3390/mps3040068

**Published:** 2020-09-30

**Authors:** Jennifer N. Gray, Theresa Wolf-Doty, Nimisha Sulejmani, Osama Gaber, David Axelrod, Basmah Abdalla, Gabriel Danovitch

**Affiliations:** 1CareDx, 3260 Bayshore Blvd, Brisbane, CA 94005, USA; twolf@caredx.com (T.W.-D.); nsulejmani@caredx.com (N.S.); 2Houston Methodist Hospital, 6565 Fannin St. Houston, TX 77030, USA; aogaber@houstonmethodist.org; 3Department of Surgery, University of Iowa Medical Center, 200 Hawkins Dr, Iowa City, IA 52242, USA; david-axelrod@uiowa.edu; 4UCLA Medical Center, 757 Westwood Plaza, Los Angeles, CA 90095, USA; BAbdalla@mednet.ucla.edu (B.A.); GDanovitch@mednet.ucla.edu (G.D.)

**Keywords:** immunominimization, renal transplant, kidney transplant, cell-free DNA, dd-cfDNA, immuno-optimization, immunosuppression, corticosteroid withdrawal, calcineurin inhibitor minimization

## Abstract

Immunosuppressant agents are essential in every transplant recipient’s care yet walking the fine line of over- or under-immunosuppression is a constant struggle for both patients and transplant providers alike. Optimization and personalization of immunosuppression has been limited by the need for non-invasive graft surveillance methods that are specific enough to identify organ injury in real time. With this in mind, we propose a pilot study protocol utilizing both donor derived cell free DNA (dd-cfDNA, gene expression profiling (GEP), and machine learning (iBox), called KidneyCare, to assess the feasibility and safety in reducing immunosuppressant exposure without increasing the risk of clinical rejection, graft injury, or allograft loss. Patients randomized to the immunominimization arm will be enrolled in one of two protocols designed to eliminate one immunosuppressant and optimize the dose of the Calcineurin Inhibitors (CNIs) using the KidneyCare platform. All patients will be maintained on dual therapy of either steroids and a low dose CNI, or mycophenolate mofetil (MMF) and low dose CNI. Their outcomes will be compared to patients who have their immunosuppressants managed using standard clinical assessment and treatment protocols to determine the impact of immuno-optimization on graft function, complications, and patient reported outcomes.

## 1. Introduction

Since 1980, cyclosporine has been administered to kidney transplant recipients (KTRs) to suppress immunologic reactions and limit allograft rejection, improving the success rate of organ transplantation. Conventional regimens today include corticosteroids, calcineurin inhibitors (CNIs), and anti-metabolite antagonists such as mycophenolate mofetil (MMF) as the most common combination of maintenance immunosuppression therapy post transplantation, although substantial variation exists [1,2]. Transplant patients continue to walk a fine line between over and under-immunosuppression. The medications that suppress the immune system also increase the likelihood of many life-threatening complications. Chronic use of steroids can cause skin thinning and poor wound healing, increase the risk of diabetes, cause mood swings/changes, and worsen osteoporosis, among other long- and short-term side effects [3]. High doses of calcineurin inhibitors, such as tacrolimus, have been found to induce nephrotoxicity that contributes to allograft loss resulting from chronic allograft injury [4]. CNIs are also associated with new onset diabetes after transplantation (NODAT) in 10–20% of patients [5]. In addition to these drug-related adverse effects, transplant recipients are also at increased risk of developing opportunistic infections and a spectrum of malignancies, many of which can be caused by viruses. Furthermore, the cost of immunosuppression related complications significantly increases the cost of post-transplant care [6]. Data on immunosuppression optimization strategies on allograft function have been promising, though unevenly adopted nationally. United Network for Organ Sharing (UNOS) data suggests up to 50% of patients are still on triple therapy two years post-transplant, with female patients more likely to be on triple therapy than male [7]. However, in some centers, 100% of patients, regardless of characteristics, were placed on triple therapy, whereas other centers used this regimen rarely if ever [7,8]. Steroid avoidance protocols are increasingly being used across North America, with studies showing that corticosteroids are not essential to achieve excellent results. However, the long-term benefit of steroid withdrawal has not been fully elucidated, with some experts asserting that early steroid withdrawal increases the risk of humoral rejection and others contending that continued use results in accelerated cardiovascular disease and osteopenia/porosis [9]. Finally, there is significant concern that underimmunosuppression predisposes patients to the development of subclinical cellular rejection and the development of donor specific antibody (DSA), which impacts on term allograft survival [10].

For this reason, the minimization of immunosuppressive agents has continued to be a controversial subject in the world of transplantation. The risk of early rejection and graft loss while altering immunosuppression renders minimization alarming to both the patient and clinician. Specifically, without a non-invasive, precise way to monitor the graft, clinicians continue to rely on typical clinical parameters that are insensitive. The degree of immune quiescence is a key determining factor when deciding whether it is appropriate to consider initiating patients on an immunominimization and optimization strategies. Unfortunately, clinical assessment and routine laboratory tests (e.g., serum creatinine, proteinuria, CNI serum trough levels) do not precisely characterize the among of ongoing, subclinical, molecular injury of the organ or changes in the recipient’s immune system in response to changing immunosuppression. These imprecise markers have been shown to be insensitive to early immunologic injury in the graft and significant findings are delayed until significant graft injury is already present [11]. While percutaneous biopsies can provide histologic assessments, repeated biopsies are not practical as a continuous monitoring strategy given its cost, invasive and painful nature, and associated complications. With the advent of dd-cfDNA, there is now a non-invasive way to monitor graft injury and thereby aid in the optimization of immunosuppression.

Safely reducing side effects of immunosuppression has been an ongoing area of interest. However, early trials have demonstrated increased rates of acute rejection, leading to clinical concern that without more effective monitoring tools, immune optimization strategies may increase the rates of graft loss, antibody formation, and long-term morbidity and mortality [8,9,12]. Current measures of allograft function, such as creatinine, remain insensitive to early rejection resulting in delayed treatment that may increase the risk of acute and chronic allograft injury (CAI) and worsen the outcome of transplant recipients managed with immunosuppression optimization strategies. In the CTOT-9 trial, the minimization of CNI and eventual removal was attempted, but the results were unfavorable, and the trial was halted [13]. The three-year follow up to the ATLAS study showed that there was a higher risk of rejection in the first six months post-transplant in the steroid free group, but that long-term outcomes, such as graft and patient survival remained similar [14].

## 2. The KidneyCare Platform-Three-Pronged, Multimodality Testing 

### 2.1. AlloSure^®^-Donor-Derived-cfDNA (dd-cfDNA)

Multiple studies have demonstrated that significant allograft injury can occur in the absence of changes in serum creatinine [10]. Thus, clinicians have limited ability to detect injury early and intervene expeditiously to prevent long term damage using creatinine alone. While biopsy and histologic analysis of the allograft remains the gold standard, it is an invasive test with enhanced risk of complications where repetitive biopsies are not well tolerated. Therefore, there are limitations in the frequent utility of biopsy to assess organ injury and differentiate rejection from other injury in kidney transplants. AlloSure^®^ (dd-cfDNA) is a non-invasive test of allograft injury that enables frequent, quantitative, and accurate assessment of allograft rejection and injury status [15,16]. Monitoring graft injury with dd-cfDNA allows clinicians to optimize the timing of allograft biopsies and guide immunosuppression management with a non-invasive and repeatable assay. Allosure^®^ also ensures response to treatment modification and resolution of graft injury.

### 2.2. AlloMap Kidney^®^-Gene Expression Profiling (GEP)

Gene expression studies have demonstrated that profiles between rejection and quiescent kidneys are different, with existing studies based on a candidate gene approach [17]. These candidate genes are mostly selected based on biologic plausibility and known polymorphisms associated with functionality (e.g., level of cytokine secretion). AlloMap Heart^®^ is a panel assay of 20 genes, 11 informative and 9 used for normalization and/or quality control, which produces gene expression data used in the calculation of an AlloMap test score-an integer ranging from 0 to 40. Compared with patients in the same post-transplant period, the lower the score, the lower the probability of acute cellular rejection at the time of testing. Likewise, AlloMap Kidney^®^ is an evolution of this platform using a proprietary learning algorithm that examines published literature for peripheral blood genes, assessing their value in kidney allograft rejection following assessment on RT-qPCR. S AlloMap Kidney^®^ is a gene panel of both informative and quality control genes, which produces gene expression data used in the calculation of an AlloMap Kidney^®^ test score-an integer ranging from 0 to 20.

### 2.3. iBox-Allograft Prognostic Score

Clinical trials evaluating the impact of immunosuppression modification on long patient outcomes has been limited by the need for extend trials of large patient cohorts. However, newly developed models which accurately predict the time to allograft failure and foresee allograft function trajectories, can now be used to predict the benefit of immune optimization. These biomarkers have been accepted by the FDA as valid clinical endpoints [18]. The independent predictors and indicators of graft prognosis include the time from transplant in years, proteinuria, eGFR (mL/min/1.73 m^2^), biopsy results to include histological findings, and the presence of HLA donor specific antibody mean fluorescence intensity [18]. Incorporating a validated and accessible scoring system into the trial protocol to assess differential rates of long-term kidney allograft failure between treatment arms allows the development of adequately powered clinical trials which can be completed in an expedited fashion.

The iBox score models determinants of allograft and patient survival (Cox model, multinomial regression), complemented with artificial intelligence and machine learning algorithms. This dynamic prediction model considers the cross-sectional aspect of variables assessed at the time of transplant, as well as longitudinal aspects of kidney function, providing a generalizable, transportable, mechanistically and data driven composite surrogate end point in kidney transplantation. iBox also provides a dynamic prediction of kidney allograft survival in the subsequent 3, 5, 7 years post-analysis as shown by the evolution over time of selected clinical and biological parameters [18] (Figure 1).

## 3. Methods

Recipients of isolated, primary kidney transplant who are 18 years of age and above and have been deemed as low immunological risk, defined as Panel Reactive Antibodies (PRA) <20%, with no donor specific antibodies (DSA) at the time of transplant will be included. Exclusion criteria include known pre-existing or *de novo* DSA, prior renal or extrarenal transplantation, eGFR < 45 mL/min/1.73 m^2^, and baseline proteinuria > 0.5 g/d, any patient on chronic steroid utilization for another indication, and any patient known to be pregnant. All patients that consent to study participation, meet all study inclusion criteria, and are not disqualified by the exclusion criteria will be randomized into the study. Consent and randomization will happen within 90 days of transplant and weaning of immunosuppressants will begin at the end of month 3, once protocol biopsy and other testing confirms it is safe to proceed. Patients willing to participate will then be randomized (1:1) into either the KidneyCare Intervention arm or the KidneyCare observational arm (Figure 2).

All centers will utilize their own hospital protocol for induction regimen at the time of transplant. Patients in the interventional arms will have stepwise decrease and elimination of either MMF or steroids based upon the monthly KidneyCare results and DSA and/or biopsy results, when applicable. Patients in the KidneyCare control arm will have a reduction and eventual elimination of either MMF or steroids based upon clinician preference using standard of care; in this arm the KidneyCare reports will not be visible to the clinical team. The weaning schedule for MMF and steroids has been outlined below with a surveillance biopsy being recommended for all study patients at 3, 12 and 24 months (Table 1, Table 2, Table 3 and Table 4).

The results of the KidneyCare draws and DSA will determine the ability to safely wean immunosuppression (Table 5). The various combinations of scores will direct clinicians as to when it is deemed safe to proceed with predetermined wean of either MMF or prednisone (Table 1 and Table 3). If the KidneyCare results reveal an elevated dd-cfDNA, new DSA, or an unfavorable iBox score, it is deemed hazardous to proceed and the immunosuppression will not be changed. The KidneyCare testing will be repeated monthly in all cases.

### Abnormal Biopsy Result

In the case of an abnormal biopsy, the patient will be treated by transplant center protocol and continuation in the protocol will be assessed by the study primary investigator. When the repeat KidneyCare draw is performed the next month and reveals it is safe to do so, the patient will then proceed with immunosuppression wean per Table 1 or Table 3, depending on designated study arm.

Study participation covers a three-year period and monthly patient participation is needed for the first two years. In total, 36 visits are required, which generally mimics the clinical standard of care and only involves blood tests. Data will be transcribed from the hospital electronic medical record (EMR) into an electronic data capture (EDC) system where key results will be transcribed from the hospital EMR into the portal within one week of the patient’s scheduled assessment visit.

## 4. Results/Conclusions

Improvements in kidney transplantation have resulted in lower acute rejection rates which has been attributed to enhanced immunosuppressive agents and their management; however, this has been offset by the rise in long-term toxicities and largely static long-term graft survival [10]. The side effects of immunosuppressant agents are unfavorable to both the patient and the allograft so mitigating these side effects is paramount to the long-term success of the organ. Safely reducing immunosuppression and thus the overall negative impact of these therapies to the organ has been an ongoing area of interest; however, early trials demonstrated increased rates of acute rejection, leading to clinical concern that without better monitoring tools, immune optimization strategies may increase the rates of graft loss, antibody formation, and long term morbidity and mortality. The risk of participating in these reduction studies is that minimization in patients who are not immunologically quiescent results in higher incidence of acute rejection and the development of DSA with long term allograft injury. The use accurate biomarker testing to monitor immune reactivity and subclinical graft has not been previously incorporated into minimization trials. The information provided by the KidneyCare Platform may provide a foundation to safely implement immunominimization and optimization protocols. KidneyCare could provide a significant advantage, as it allows clinicians to identify previously unrecognized injury while optimizing the beneficial immunosuppression needed to keep their kidney transplant healthy. Early recognition of subclinical rejection and graft injury in, allows intervention in the subset of patients who require enhanced immunosuppression prior to irreversible injury. While the benefits of appropriate immunominimization complications and cost are clear, we believe that the risk of immunosuppression reduction can be mitigated by only lowering medication doses when the molecular KidneyCare platform suggests a state of immune-quiescence and no organ injury.

## Figures and Tables

**Figure 1 mps-03-00068-f001:**
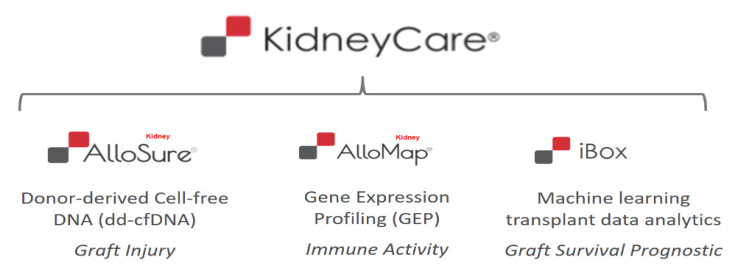
Multimodality testing using dd-cfDNA, GEP and machine learning to better assess organ health and enable the optimization of immunosuppression.

**Figure 2 mps-03-00068-f002:**
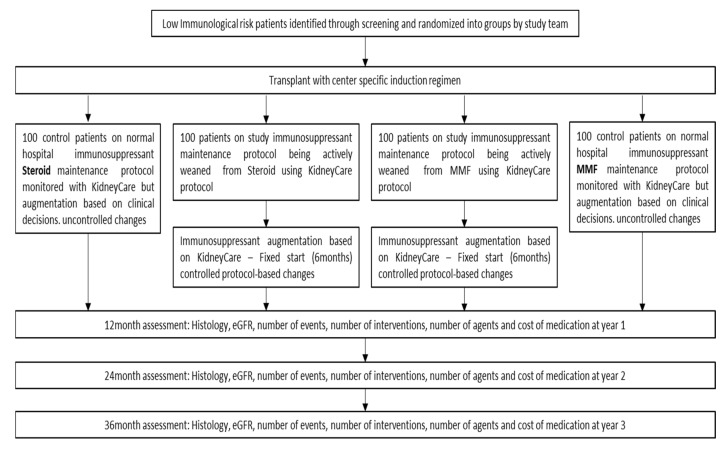
KIRA study design using multimodality testing to optimize immunosuppression.

**Table 1 mps-03-00068-t001:** Unblinded, two-armed interventional trial–MMF minimization arm, first year post transplant.

Post-Tx Regimen	Week 1	Mon 1	Mon 2	Mon 3	Mon 4	Mon 5	Mon 6	Mon 7	Mon 8	Mon 9	Mon 10	Mon 11	Mon 12
Prednisone	Center protocol	5 mg	5 mg	5 mg	5 mg	5 mg	5 mg	5 mg	5 mg	5 mg	5 mg	5 mg	5 mg
MMF q 12 h	1000 mg	1000 mg	1000 mg	1000 mg	750 mg	750 mg	500 mg	500 mg	500 mg	250 mg	250 mg	0 mg	0 mg
Tacrolimus goal (ng/mL)	8–12	8–12	8–12	8–12	8–12	8–12	8–12	6–8	6–8	6–8	6–8	6–8	6–8
DSA	X			X			X			X			X
BK screening		X	X	X	X	X	X	X	X	X	X	X	X
KidneyCare	X	X	X	X	X	X	X	X	X	X	X	X	X
Protocol Bx				X									X

**Table 2 mps-03-00068-t002:** Year Two-Continue Immunosuppressant Regimen from Month 12.

Post Tx Regimen	13 Month	14	15	16	17	18	19	20	21	22	23	24
DSA			X			X			X			X
BK screening	X	X	X	X	X	X	X	X	X	X	X	X
KidneyCare	X	X	X	X	X	X	X	X	X	X	X	X
Cystatin C						X						X
Protocol Bx												X

**Table 3 mps-03-00068-t003:** Unblinded, two-armed interventional trial- Steroid minimization arm, first year post transplant.

Post Tx Regimen	Week 1	Mon 1	Mon 2	Mon 3	Mon 4	Mon 5	Mon 6	Mon 7	Mon 8	Mon 9	Mon 10	Mon 11	Mon 12
Prednisone	30 mg	10 mg	5 mg	5 mg	0 mg	0 mg	0 mg	0 mg	0 mg	0 mg	0 mg	0 mg	0 mg
MMF q 12 h	1000 mg	1000 mg	1000 mg	1000 mg	750 mg	750 mg	500 mg	500 mg	500 mg	500 mg	500 mg	500 mg	500 mg
Tacrolimus goal (ng/mL)	8–12	8–12	8–12	8–11	8–12	8–12	8–12	8–12	8–12	6–8	6–8	6–8	6–8
DSA	X			X			X			X			X
BK screening		X	X	X	X	X	X	X	X	X	X	X	X
Kidney Care	X	X	X	X	X	X	X	X	X	X	X	X	X
Protocol Bx				X									X

**Table 4 mps-03-00068-t004:** Year Two-Continue Immunosuppressant Regimen from Month 12.

Post Tx Regimen	13 Month	14	15	16	17	18	19	20	21	22	23	24
DSA			X			X			X			X
BK screening	X	X	X	X	X	X	X	X	X	X	X	X
KidneyCare	X	X	X	X	X	X	X	X	X	X	X	X
Cystatin C						X						X
Protocol Bx												X

**Table 5 mps-03-00068-t005:** Immunosuppression minimization/optimization will be determined by KidneyCare Results and DSA as to when it is safe to proceed.

New DSA	AlloSure (%)	3 yr iBox Score	AlloMap Kidney (RUO)	Action
Negative	≤0.2	> 85%	<10	Proceed/Minimize
Negative	>0.2 ≤ 0.5	>85%	<10	Proceed/Minimize
Negative	>0.5 + ≤ 61% change	>85%	<10	Proceed/Minimize
Positive	>0.2 ≤ 0.5	<85%	Any value	Repeat
Negative	>0.5 + >61% change	>85%	Any value	Repeat
Positive	>0.5 + ≤61% change	<85%	Any value	Not Safe to Proceed
Positive	>0.5 + >61% change	<85%	Any value	Not Safe to Proceed
Positive	≤0.2	<85%	>10	Not Safe to proceed
Positive	>1	<85%	Any value	Not Safe to Proceed
Negative	>1	>85%	Any value	Not Safe to Proceed

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
