# Peer review of "KidneyCare Guided Immuno-Optimization in Renal Allografts: The KIRA Protocol"

_mps, 2020, doi:10.3390/mps3040068_

Round 1

Reviewer 1 Report

Gray and coworkers presented an interesting study protocol: Kidney Care Guided Immuno-optimization in Renal Allografts: The KIRA protocol.

The authors presented a study protocol, used appropriate methodology, correctly emphasized some of the strengths and limitations, and finally the expectations.

Closer look at the presented manuscript have raised some comments.

Comments:

  1. The authors should explain more clearly multimodality testing and variables used iBox score models.
  2. Authors should define the eGFR method more precisely.
  3. Some abbreviations need explanations, like BK, Bx, UNOS, DBD, DCD, EMR…

The authors should accept and discuss these comments in the manuscript.

Author Response

  1. The authors should explain more clearly multimodality testing and variables used iBox score models. 

Thank you for your comments.  The 9 parameters of the iBox algorithm have been added to the manuscript and the citation is #18

2. Authors should define the eGFR method more precisely.

Clayton et al concluded that 30% decline in eGFR between years 1 and 3 after kidney transplant is common and strongly associated with risks of subsequent death and death–censored graft failure, which mirrors findings in CKD. Percentage decline in eGFR should be considered for use as a surrogate outcome in kidney transplant trials, hence we will consider the %difference in decline between groups.

3. Some abbreviations need explanations, like BK, Bx, UNOS, DBD, DCD, EMR…

Thank you -- these have been written out and included explanations where applicable. 

Reviewer 2 Report

In the manuscript the authors present a protocol to be executed in order to decrease the immunosuppressive drugs used in patients receiving a renal transplant. In order to do this, they will evaluate the potentiality of rejection according to the results obtained in the commercial platform of kidney care. The potential adverse effects of the overuse of immunosuppressive drugs are described, as well as the possibility of rejection in the underuse of these drugs. The reduction of steroids and mycophenolate mofetil is proposed according to the results of the kidney care platform.

The proposed objective and protocol are adequate, but the work lacks some important points.

-It should be described in much more detail what each of the three platform tools consists of:

1.- On which is based the detection of circulating donor DNA,( SNP, HLA ... ) To be clarified.

2.- What are the main genes studied in the AlloMap Kidney®?

3.- What parameters does iBox Allograft Prognostic Score collect?

-The results obtained from the three items of the platform should be described, how they are and how they are quantified. If they are not described it is very complicated to assess how these parameters will affect the protocol

-On the other hand, it should be described how the results obtained from the platform will affect decisions on the reduction of immunosuppressants. From which results the reduction of immunosuppressors is suspended. For how long?

Author Response

1.- On which is based the detection of circulating donor DNA,( SNP, HLA ... ) To be clarified.

AlloSure uses NGS SNP based technology – the analytical validation of this method is cited in the text as a reference 12.

2.- What are the main genes studied in the AlloMap Kidney®?

AlloMap kidney uses a propriety selection of genes, where the full panel is not yet disclosed. Publicly it has been announced that the panel used within AlloMap Heart in addition to the top 50 genes identified in the a machine learning search of the literature; have been added to the panel.

3.- What parameters does iBox Allograft Prognostic Score collect? 

Thank you for the guidance. The validation paper for ibox has been cited (18). The full number of parameters are listed within this BMJ article and so will be easily accessible, an overview of the 9 parameters used has been added to the manuscript.  

4.- On the other hand, it should be described how the results obtained from the platform will affect decisions on the reduction of immunosuppressants. From which results the reduction of immunosuppressors is suspended. For how long?

Please see attachment -- this has been added to the manuscript.

Round 2

Reviewer 2 Report

The suggestions I proposed to the manuscript have been made. Under these conditions the article is understandable.